# Self-Template Synthesis of Nitrogen-Doped Hollow Carbon Nanospheres with Rational Mesoporosity for Efficient Supercapacitors

**DOI:** 10.3390/ma14133619

**Published:** 2021-06-29

**Authors:** Xiang Zhao, Mu Zhang, Wei Pan, Rui Yang, Xudong Sun

**Affiliations:** 1Key Laboratory for Anisotropy and Texture of Materials (Ministry of Education), Northeastern University, Shenyang 110819, China; 1310142@stu.neu.edu.cn (X.Z.); 1510139@stu.neu.edu.cn (W.P.); yangrui@stumail.neu.edu.cn (R.Y.); 2Lab. of Advanced Ceramics, Foshan Graduate School of Northeastern University, Foshan 528311, China

**Keywords:** self-template, hollow structure, multidirectional porosity, nitrogen-doped carbon, supercapacitors

## Abstract

Rational design and economic fabrication are essential to develop carbonic electrode materials with optimized porosity for high-performance supercapacitors. Herein, nitrogen-doped hollow carbon nanospheres (NHCSs) derived from resorcinol and formaldehyde resin are successfully prepared via a self-template strategy. The porosity and heteroatoms in the carbon shell can be adjusted by purposefully introducing various dosages of ammonium ferric citrate (AFC). Under the optimum AFC dosage (30 mg), the as-prepared NHCS-30 possesses hierarchical architecture, high specific surface area up to 1987 m^2^·g^−1^, an ultrahigh mesopore proportion of 98%, and moderate contents of heteroatoms, and these features endow it with a high specific capacitance of 206.5 F·g^−1^ at 0.2 A·g^−1^, with a good rate capability of 125 F·g^−1^ at 20 A·g^−1^ as well as outstanding electrochemical stability after 5000 cycles in a 6 M KOH electrolyte. Furthermore, the assembled NHCS-30 based symmetric supercapacitor delivers an energy density of 14.1 W·h·kg^−1^ at a power density of 200 W·kg^−1^ in a 6 M KOH electrolyte. This work provides not only an appealing model to study the effect of structural and component change on capacitance, but also general guidance to expand functionality electrode materials by the self-template method.

## 1. Introduction

Up to now, electrical energy has been the most widespread and convenient form of energy usage, and the consumption of electricity will surge with the rising population [1,2,3]. These ever-increasing demands are expected to be derived from renewable sources like solar, tidal, wind, and biomass power etc. [4,5,6,7], with the aim of reducing fossil fuel depletion and greenhouse gas emission. However, the direct penetration of the converted energy into the electrical grid shows a non-linear output characteristic due to the variable and intermittent nature. Such a knotty problem poses a challenge to develop stable and high-energy density energy storage devices as a transfer station [8,9,10,11,12]. Among varied candidates, supercapacitors (SCs) that respond rapidly to the discontinuous changes from the conversion of renewable energy have potential to deliver large energy densities, stably at high rates.

Electrochemical double layer capacitors (EDLCs) as a representative SC have characteristics of high-power density, superior durability, and abundant resources [13,14]. In EDLCs, electric charges are stored through reversible ion adsorption/desorption in the pores of carbon material or on the electrode/electrolyte interface. As a result, capacitive behavior is strongly dependent on the surface area and the porosity of the carbolic electrodes. Expanding the specific surface area (SSA) of the carbon material contributes to the charge arrangement, while rational allocation of pore structure improves ion transfer efficiency [15,16]. In addition, heterogeneous doping that improves the affinity of the electrolyte and generates extra pseudocapacitance is common as a competing means to enhance capacitive behavior [17]. These promising improvements of EDLCs have pointed to an optimized model that possesses hierarchical architecture and reasonable composition.

According to the model, nitrogen-doped hollow carbon nanospheres (NHCSs) show unique structural advantages and have gained attention in the energy storage fields [18,19]. In comparison with single linear [12], flaky [20,21], and bulk [22] structure, a hollow structure reasonably combines the advantages of micro-, meso-, and macropores, which actually act as extra sources of faradaic pseudocapacitance, a highway for ion transport, and a buffering reservoir for electrolyte, respectively, and thus exhibits superior performance in energy storage areas [23]. Meanwhile, an efficient nitrogenous dopant can increase the electrical conductivity, hydrophilicity, and the surface basic sites of NHCSs, which is beneficial for the utilization of active sites and a reduction of impedance in aqueous electrolyte [24,25,26,27].

In general, the synthesis of NHCSs is restricted to a template-based strategy, mainly including the templated core establishment and layer-by-layer covering. However, these processes need to be improved as the post-treatment for template removal is harsh and time consuming, during which the structural collapse results in the decrease of the specific surface area and the number of accessible mesopores, while inevitably shielding some active sites [28,29,30]. In addition, appropriate heteroatomic doping and targeted porous adjustment engineering to enhance the inherent physicochemical properties and the heterogeneous interfaces of the carbon scaffolds are hard to realize simultaneously in most cases. It is worth noting that activated carbon obtained from biomass compounds can also provide a hierarchically porous structure and inherent heteroatoms [31,32,33,34]. Admittedly these materials have achieved high specific capacitance but what is missing in these researches is that most of them are opportunistic with lack of systematization, while the basis of the research target selection and the further modification method are seldom reported. Thus, the choice of a straightforward synthetic method can not only realize the high capacitance characteristic, but also meet the quality control of the electrode materials.

The use of resorcinol-formaldehyde (RF) resin as a carbon source for NHCSs is a fascinating strategy for satisfying the needs of scientific-commercial goals. The multiple synthetic routes that occur in both acidic and alkaline conditions present similar polymerization to the hydrolysis of organosilicon compounds [35,36,37]. Recently, our group explored a new path to synthesize nitrogen-doped resorcinol-formaldehyde spheres (N-RFS) with precisely tailored size control [38]. According to the new-found “outside-in” growth pattern, the inner intermediates can be selectively dissolved by acetone, which opens up new design space by creating a cavity in-situ without a variable template and an extra enucleation step. At the same time, the porous structure and composition can be regulated by nitrogen-rich structure-directing agent polymerized with the functionalized active sites of resorcinol [39,40], having the advantages of saving reaction time, a flexible adjustment approach, high-yield micropores, and high SSA.

In this paper, a novel self-template strategy is proposed for the synthesis of nitrogen-doped hollow carbon nanospheres (NHCSs). The polymeric precursor is produced via polymerization between resorcinol and formaldehyde in an alkaline solution, and subsequent “enucleation process”. The enucleation process involves two main steps: one is a combined effect of acetone and CTAB, during which the continuously dissolved interior intermediates are assembled on the residual shell; and the other is the introduction of ammonium ferric citrate (AFC), during which adjustable mesopores and heteroatom atoms are successfully formed in the skeleton. After a mild heat-treatment process, the products as an alternative capacitive material were systematically investigated and have potential as an ideal model to study the effects of structural change and heteroatom doping on capacitive behavior.

## 2. Materials and Methods

### 2.1. Materials

All chemicals used in the experiments were of analytical grade without further purification. Among which, resorcinol, formaldehyde solution (37 wt.%), methanol, ammonium iron citrate (AFC), and ammonium hydroxide solution (28 wt.%) were purchased from Shanghai Aladdin Biochemical Technology Co., Ltd. (Shanghai, China). Cetyltrimethylammonium bromide (CTAB) and acetone were purchased from Sinopharm Chemical Reagent Co., Ltd. (Beijing, China).

### 2.2. Preparation of Nitrogen-Doped Hollow Carbon Nanospheres

Nitrogen-doped hollow carbon nanospheres (NHCSs) were synthesized through an innovative self-template method. Namely, resorcinol (0.551 g) and formaldehyde (0.72 mL) first underwent polycondensation with ammonium hydroxide solution (1.15 mL) as adscititious alkali in a 50 mL mixed solvent (10 vol% methanol in deionized water) for 6 h. Then, 0.4 g of CTAB and 25 mL of acetone were added successively and stirred until a nearly transparent solution occurred. Next, different dosages of AFC, e.g., 0, 10, 30, and 50 mg were added and stirred for a further 12 h. Finally, the chestnut solid product was collected by centrifugation and carbonized at 800 °C for 2 h in an argon atmosphere with a ramping rate of 2 °C·min^−1^. The products were labeled as NHCS-m, where “NHCS” refers to nitrogen-doped hollow carbon nanospheres, and the symbol “m” represents the AFC dosage used (in mg).

### 2.3. Characterization

Field emission scanning electron microscope (FE-SEM), transmission electron microscope (TEM) and energy dispersive spectrometer (EDS) images were obtained using Hitachi S-4800 (Tokyo, Japan) and FEI Tecnai G2 F20 S-Twin (Hillsboro, OR, USA), respectively. The nitrogen adsorption/desorption isotherms were measured on a Quantachrome nitrogen adsorption instrument. The specific surface area (SSA) of the samples was calculated by the Brunauer–Emmett-Teller (BET) method. The pore size distribution (PSD) was calculated from the adsorption isotherms by the density functional theory (DFT). Powder X-ray diffraction (XRD) spectra were recorded with a Rigaku (Tokyo, Japan) Ultima IV with Cu Kα radiation. Raman spectra were collected using a HORIBA Scientific XploRA PLUS Raman spectrometer. X-ray photoelectron spectroscopy (XPS) measurements were performed on an Axis Supra using a monochromic Al Kα X-ray radiation.

### 2.4. Electrochemical Measurements

The working electrode was constructed as follows: the NHCS-*m* samples and polyvinylidene fluoride (PVDF) binder (90:10 weight ratio) were dispersed in *N*-methyl-2-pyrrolidinone (NMP). Then, the suspension was pressed onto a nickel foam (10 mm × 10 mm) at 10 MPa and dried at 60 °C in a vacuum oven for 12 h. The mass loading of NHCS-*m* in each working electrode was ~2 mg. Electrochemical measurements of the working electrode were first conducted in a three-electrode cell, including 6 M KOH as aqueous electrolyte, a platinum foil as the counter electrode, and a saturated calomel electrode (SCE) as the reference electrode, respectively, and then characterized by cyclic voltammetry (CV), galvanostatic charge/discharge (GCD), and electrochemical impedance spectroscopy (EIS) on a Zahner Ennium E electrochemical workstation. As for the two-electrode cell, the NHCS-30 based symmetric supercapacitors were assembled by two identical working electrodes in 6 M KOH with one piece of cellulose paper (Whatman (Little Chalfont, UK), GF/D) as the separator.

For the three-electrode cell, the specific capacitance was calculated by CV and GCD curves, respectively, according to the following Equations (1) and (2):(1)$$C_{s}=\frac{\int_{V_{1}}^{V_{2}}I\left(V\right)dV}{m\upsilon \left(V_{2}-V_{1}\right)}$$
(2)$$C_{p}=\frac{I∆t}{m∆V}$$
where $C_{s}$ and $C_{p}$ are the specific capacitances obtained from CV and GCD curves, $V_{1}$ and $V_{2}$ are the low and high potential limits in the CV curves, *I* (V), *m* (g), *υ* (mV·s^−1^), ∆t (s), ∆V (V) refer to the instant current, mass loading, scan rate, discharge time, and potential window, respectively.

For the two-electrode cell, the specific capacitance was also calculated according to Equation (2). The difference is that *m* (g) refers to the whole mass loading of active materials. The energy density (*E*, W·h·kg^−1^) and power density (*P*, W·kg^−1^) were calculated respectively according to the following Equations (3) and (4):(3)$$E=\frac{0.5C{\left(∆V\right)}^{2}}{3.6}$$
(4)$$P=\frac{3600E}{∆t}$$
where *C* (F·g^−1^) is the capacitance of the cell, ∆V (V) is the discharge voltage range, ∆t (s) is the discharge time.

## 3. Results and Discussion

The strategy for synthesizing nitrogen-doped hollow carbon nanospheres (NHCSs) is illustrated in Figure 1, mainly comprised of template generation and new shell establishment. First, large RF microemulsions were formed due to the polymerization between resorcinol and formaldehyde in the presence of an appropriate amount of ammonia solution. Subsequently, the newly formed porous shell could be explained by a step-by-step process including the orientation arrangement of CTAB, selective dissolution of acetone, deprotonation between AFC and intermediates, and subsequent condensation polymerization. Specifically, the newly added CTAB was liable to be anchored on the surface of the RF emulsions due to the electronic interaction between the positively charged CTAB hydrophobic side and the negatively charged phenolic hydroxyl group so that an outmost oriented adsorbed layer was formed. With the introduction of acetone, the interior non-solidified RF intermediates were selectively dissolved, passed through the shell, and ultimately absorbed by CTAB. Further, deprotonation continued to occur between the adsorbed intermediates and AFC, resulting in the formation of a denser RF/AFC composite layer. Thereby, uniform nitrogen-doped hollow RF nanospheres (denoted as NHRFSs) were successfully fabricated and transformed into NHCSs after calcination at 800 °C for 2 h in argon. Based on the color change of the reaction system from milky white (RF emulsions) to translucence (RF hollow spheres), then to chestnut (NHRFSs), and finally to black (NHCSs) at various stages of the process, a general reaction process may be deduced.

Based on the above synthetic route, it can be speculated that the dosage of AFC plays an important role in regulating the architecture of the products on account of the competitive system between the deprotonation of AFC and the polycondensation of formaldehyde with resorcinol under alkaline conditions. Field-emission scanning electron microscopy (FE-SEM) and Transmission electron microscopy (TEM) were first used to investigate the morphology and structure changes of NHRFS-*m* and corresponding NHCS-*m* samples. The FE-SEM images (Appendix A) clearly show that the spherical NHRFS-*m* samples have similar surface roughness and good dispersion. In addition, the average particle sizes are 250.2, 246.1, 245.7, and 245.1 nm for NHRFS-0, NHRFS-10, NHRFS-30, and NHRFS-50, respectively. The observed results show that the geometry and size of the NHRFS-*m* samples are not distinctly altered by the changes of AFC dosage. After the carbonation process, the corresponding NHCS-*m* samples are still uniform and slightly contractive, and the average sizes are 203.2, 193.3, 191.3, and 187.7 nm for NHCS-0, NHCS-10, NHCS-30, and NHCS-50, respectively (Figure 2a–d). Meanwhile, a large number of crater-like pores via organic decomposition are disorderly distributed throughout the carbon shells (Figure 2e–h). A careful observation of the TEM images (Figure 2i–l) shows that the outer pores in the NHCS-*m* sample are connect channels from the cavity to the outside. Without AFC addition, polycondensation of formaldehyde with resorcinol took the dominant role, and the generated intermediates continued to grow along the CTAB chains, thus leading to the formation of center-radial mesoporous channels (Figure 2m). With less AFC dosage (<30 mg), narrow and curved channels were observed, indicating that the deprotonation of AFC was involved in the formation of intermediates, and as a result, the grafted AFC chains were beneficial in increasing the degree of crosslinking and spatial confusion (Figure 2n,o). On increasing the dosage of AFC to 50 mg, the carbon shell showed an irregular channel morphology due to the dominant AFC deprotonation and the resulting complex intermediates (Figure 2p). The hollow structure was further supported by elemental mapping from TEM-energy dispersive spectroscopy (EDS). EDS results of an individual NHCS-30 nanosphere revealed the presence and homogenous distribution of C, Fe, N, and O elements (Appendix A). 

The textural properties of NHCS-*m* were further analyzed by the nitrogen adsorption technique. The nitrogen adsorption/desorption isotherm and the corresponding pore size distribution curves calculated by the density functional theory (DFT) method are shown in Figure 3a,b, respectively. As shown in Figure 3a, all samples exhibit type-IV isotherm curves with two steep uptakes (*P*/*P_0_* < 0.01, *P*/*P_0_* > 0.96) and an obvious hysteresis loop (0.4 < *P*/*P_0_* < 0.96), suggesting a hierarchically skeletal structure and coexistence of micropores, mesopores, and macropores [41]. The micro-, meso-, and macropores are considered to act as an extra source of faradaic pseudocapacitance, a highway for ion transport, and a buffering reservoir for electrolyte, respectively. The pore size distribution curves shown in Figure 3b exhibit a similar pore width (5.7–7.1 nm) in all samples. The detailed SSA and pore structure parameters for the NHCS-*m* samples are summarized in Table 1. It can be noted that the total pore volume decreases apparently from 5.37 to 2.33 cm^3^·g^−1^ with the increase of AFC dosage, while there is no significant change in micropore volumes, demonstrating the negative effect of the grafted AFC chains on making mesopores. The BET specific surface areas of NHCS-0, NHCS-10, NHCS-30, and NHCS-50 were 1433, 1513, 1987, and 1745 m^2^·g^−1^, respectively, showing an opposite trend to the total pore volume. The decreased SSA of NHCS-50 demonstrated the generation of closed pores with the excessive AFC dosage and the limited and targeted structural adjustment of AFC.

As reported previously, not only rational structure design, but also the incorporation of heteroatoms plays an important role in affecting the electrochemical performance of the carbon-based electrodes. Therefore, X-ray diffraction (XRD) and Raman spectra were used to analyze the evolution on the phase composition and crystal structure of the NHCS-*m* samples with increased AFC dosage. As shown in Figure 3c, two common diffraction peaks at around 25.4° and 43.9° correspond to the (002) and (100) lattice planes of graphitic carbon (PDF# 41-1487), respectively [42]. On further increasing the AFC dosage to 50 mg, new characteristic peaks were observed at 30.1°, 35.4°, 43.1°, 53.1°, 56.9°, and 62.5° and were assigned to the (220), (311), (400), (422), (511), and (440) lattice planes, respectively, of Fe_3_O_4_ (PDF# 19-0629) [38], indicating that the extra Fe atoms existed in the NHCS-*m* samples as iron oxide. Figure 3d shows that two Raman peaks located at 1350 and 1590 cm^−1^ can be regarded as D band (1350 cm^−1^, the disorder in the sp^2^ carbon network) and G band (1590 cm^−1^, the sp^2^ carbon atom vibrations along the structure axis) of NHCS-*m* [43]. The intensity ratio (*I*_D_/*I*_G_) of these two peaks reflects the graphitization degree of the samples, which is an institutionalized indicator used to estimate the electroconductibility [44]. By detailed peak-fitting-analysis (Appendix A), the *I*_D_/*I*_G_ ratios for NHCS- 0, NHCS-10, NHCS-30, and NHCS-50 were calculated to be 1.10, 1.08, 1.02, and 0.98, respectively. The gradually decreased intensity of D band for the NHCS-*m* samples implies that more amorphous carbon has been converted to graphitized carbon with the increase of the AFC dosage, which originates from the catalytic activity of derived Fe_3_O_4_ at high temperature [45].

To ulteriorly clarify the surface elemental composition and atomic configurations for NHCS-30, X-ray photoelectron spectroscopy (XPS) measurements were conducted. As shown in Figure 4a, the survey spectrum of NHCS-30 contains C 1s, N 1s, O 1s, and Fe 2p. The C 1s peaks at 284.6, 286.1, 287.2, and 289.6 eV are assigned to C–C coordination, C– OH linkage, carbonyl groups, and ester groups, respectively (Figure 4b) [46]. The Fe 2p spectrum is fitted into four peaks (Figure 4c). Specifically, the peaks at 707.6 and 713.0 eV are assigned to Fe 2p_3/2_, and the other two at 720.2 and 726.0 eV are assigned to Fe 2p_1/2_, representing the co-existence of Fe^2+^ and Fe^3+^ [47]. The N 1s spectrum is deconvolved into four peaks centered at 397.7, 399.2, 400.4, and 401.8 eV, corresponding to pyridinic N, pyrrolic N, graphitic N, and oxidized N, respectively (Figure 4d) [48]. The O 1s peaks reveal the presence of carbonyl (530.8 eV), ether linkage (531.7 eV), carboxyl (533.0 eV), and N–O bond (534.2 eV) (Figure 4e) [49]. These results confirm that only nitrogen and oxygen elements are effectively doped into the carbon matrix, and the concomitant iron element is evenly distributed in the form of oxide. In addition, the contents of Fe, N, and O increase as the AFC dosage increases (Figure 4f). It has been well-documented that both the heteroatom and the oxide have proved to be electrochemical active sites, which can enhance the wettability of the electrode and provide extra pseudocapacitance.

Based on the above analysis, the as-prepared NHCS-*m* samples with high SSA, rational hierarchical structure and rich component configuration are anticipated to show satisfactory electrochemical performance for supercapacitors.

The electrochemical performance of the NHCS-*m* electrodes was first evaluated by CV, GCD, EIS, and the cycling stability test in a three-electrode system. The CV curves of all the NHCS-*m* electrodes were measured at scan rates from 2 to 200 mV·s^−1^, which possess quasi-rectangular shape and enclosed profile (Appendix A). In each electrode sample, the enlarged CV curves remain quasi-rectangular shape with the increase of scan rate, indicating a high-power behavior. Appendix A compares the CV curves of Ni foam and the NHCS-*m* electrodes at a scan rate of 5 mV·s^−1^. A pair of redox peaks can be clearly observed at −0.5 V and −0.35 V due to the combined effect of Faradaic reactions from N (pyrrolic N and pyridinic N), O (–COOH and –OH), and iron species (Fe^3+^/Fe^2+^) in the alkaline electrolyte [50,51]. Besides, the quasi-rectangular integral area increased first and then decreased with the rise of the AFC dosage. With 30 mg of the AFC dosage, the integral area was the largest. This trend is the same as that of the BET analysis, implying that the energy storage behavior of NHCS-*m* primarily relies on structural change, highlighting the role of AFC in creating high SSA.

As can been seen in Figure 5a, the specific capacitances were calculated by the CV curves obtained at various scan rates. NHCS-30 achieves the highest specific capacitance (*C*s) of 205.3 F·g^−1^ at 2 mV·s^−1^, even maintains 124.9 F·g^−1^ at a hundredfold scan rate (200 mV·s^−1^), followed by NHCS-50 (168.5 F·g^−1^ at 2 mV·s^−1^, and 101.9 F·g^−1^ at 200 mV·s^−1^), NHCS-10 (164.2 F·g^−1^ at 2 mV·s^−1^, and 97.6 F·g^−1^ at 200 mV·s^−1^), and finally NHCS-0 (93.8 F·g^−1^ at 2 mV·s^−1^, and 58.2 F·g^−1^ at 200 mV·s^−1^). Considering the similar micropore volume and ever-rising content of heteroatoms in the NHCS-*m* samples, the root cause of the highest capacitance of NHCS-30 lies in the highest SSA (1987 m^2^·g^−1^) and mesopore proportion (98%), which are conductive to the double layer formation and the ion transport, respectively.

The total capacitance of the NHCS-*m* electrodes can be divided into two parts: a surface-induced capacitive process (regarded as the fast kinetics part) and a diffusion-limited capacitive process (regarded as the slow kinetics part). The surface-induced capacitive process operates mainly through reversible ion adsorption/desorption at the electrode/electrolyte interface. While the diffusion-limited capacitive process results from faradaic redox reactions from heteroatomic states and the electrostatic accumulation in the micropores. The capacitance component was first qualitatively analyzed by the following Equations (5) and (6) [52]:(5)$$i_{p}=a\upsilon^{b}$$
(6)$$\log\left(i_{p}\right)=\log\left(a\right)+blog\left(\upsilon \right)$$
where *i*_p_ is the charge/discharge peak current, *υ* is scan rate, *a* and *b* are adjustable parameters. Particularly, the calculation result as *b* = 1 stands for the surface-induced capacitive process while *b* = 0.5 represents the diffusion-limited capacitive process [53]. As depicted in Figure 5b, well-defined fitted straight lines can be observed for NHCS-30 and the calculated *b* values (0.93 and 0.92) demonstrate the predominant kinetics of a surface-induced capacitive process. To analyze further quantitatively the capacitance component, the following Equations (7) and (8) were employed [54,55]:(7)$$i_{p}=k_{1}\upsilon +k_{2}\upsilon^{1/2}$$
(8)$$\frac{i\left(V\right)}{\upsilon^{1/2}}=k_{1}\upsilon^{1/2}+k_{2}$$
where *k*_1_ and *k*_2_ are proportionality constants related to the surface-induced capacitive process and diffusion-limited capacitive process. As shown in Figure 5c, the fitting result of NHCS-30 at 20 mV·s^−1^ presents ~66% shaded region derived from the surface-induced capacitive process. The capacitance component comparison at various scan rates is summarized in Figure 5d, and the results show that as the scan rate increases, the surface-induced capacitors (the orange part) remain constant while the contribution of the diffusion-limited capacitors (the green part) decreases from 42.22% to 6.14%. Similar results for the other three electrodes are shown in Appendix A. This occurred due to a time-limited diffusion of electrolyte ions into electrode pores at high scan rates, leading to a difficult charge arrangement in the micropores and the lacking redox reactions of the heteroatoms.

A set of Galvanostatic charge/discharge (GCD) curves of the NHCS-*m* electrodes was measured at different current densities, ranging from 0.2 to 20 A·g^−1^ between −0.8 to 0 V (Appendix A). All the GCD curves exhibited an approximate isosceles triangle, representing favorable electrochemical reversibility. Moreover, the linear behavior and slight asymmetry implied that the dominant electrical double-layer capacitance (EDLC) coexisted with the pseudocapacitance, this was consistent with the CV results. A negligible voltage drop could also be observed in each discharge curve, which arose from the sum of solution resistance and internal resistance. The higher the AFC dosage, the smaller the voltage drop, the easier the mass transfer, and the less energy loss. NHCS-30 exhibited the longest discharging time 112 s at 1 A·g^−1^, which was much superior to NHCS-0 (73 s), NHCS-10 (85 s), and NHCS-50 (107 s), corresponding to the capacitance sorting (Figure 6a). Next, the gravimetric capacitance values *C*_p_ calculated by the discharging time vs. current densities of NHCS-m are shown in Figure 6b. The *C*_p_ values obtained at 0.2 A·g^−1^ of NHCS-0, NHCS-10, NHCS-30, and NHCS-50 were 106.8, 131.8, 206.5, and 161.5 F·g^−1^, respectively, and they still exhibited high capacitances of 75, 79, 125, and 120 F·g^−1^ at a high loading current of 20 A·g^−1^, showing a satisfactory endurance and a good rate capability at a large current. Among the series of the NHCS-m electrodes, the *C*_p_ of the NHCS-30 electrode was the largest. The results confirmed that the *C*_p_ is related to SSA, and larger SSA can provide more electrode/electrolyte contact area. Long-term cycling performance is also considered as an important parameter for the electrode materials. Figure 6c shows the specific capacitance variations for the NHCS-m electrodes as a function of cycle number at a current density of 5 A·g^−1^. There were about 84.6%, 73.3%, 85.0%, and 89.5% specific capacitance retentions for NHCS-0, NHCS-10, NHCS-30, and NHCS-50, respectively, after 5000 cycles, showing good cycle stability.

To better understand the ion diffusion and charge transfer behaviors of the NHCS-*m* electrodes, electrochemical impedance spectroscopy (EIS) measurements were carried out at a frequency range from 10^−2^ to 10^5^ Hz with amplitude voltage of 5 mV. The Nyquist plots consist of a small semicircle in the high-frequency region, a near-vertical line in the low-frequency region, and a slope of 45° connection part (Warburg diffusion line), which are the characteristics of a joint pseudocapacitance from the redox process, the capacitance behavior, and a diffusion-controlled process (Figure 6d) [56]. The semicircle in the high-frequency region is associated with the charge-transfer resistance (Rct), while NHCS-30 exhibits a minimal radius in all NHCS-m, demonstrating the most efficient charge transfer between the interfaces of the KOH electrolyte and the NHCS-30 electrode. The intersection of the Nyquist plot and the real axis reflects the equivalent series resistance (Rs). Values of Rs were calculated to be 0.24, 0.19, 0.20, and 0.44 Ω for NHCS-0, NHCS-10, NHCS- 30, and NHCS-50, respectively. The decreased Rs can be attributed to the increased content of heteroatoms and electroconductivity. However, surplus AFC dosage introduced a higher intrinsic resistance of Fe_3_O_4_, which had a negative impact on the Rs, NHCS-50 in this case. More importantly, the steepest slope of the vertical line at the low-frequency region and the shortest Warburg diffusion line compared to the counterparts indicate that NHCS-30 possesses the best capacitance behavior and ion-diffusion efficiency, which can be attributed to the ultrahigh mesopore proportion for the unimpeded channels of ion diffusion. Through the above comparison, the faster ion diffusion and charge transfer endow the NHCS-30 electrode with superior electrochemical performance.

To finally explore the practical application of the as-prepared NHCS-30 electrode, a symmetric supercapacitor was assembled of two NHCS-30 electrodes as both positive and negative electrodes in a 6 M KOH electrolyte. As shown in Figure 7a, the CV curves exhibit approximately rectangular profiles at different scan rates, suggesting fast ion diffusion and charge propagation in the NHCS-30 electrodes. The rectangular area broadens with a scan rate from 2 to 200 mV·s^−1^, but the profile is not distorted, suggesting the good reversibility in the KOH electrolyte. The GCD curves measured from 0.5 to 10 A·g^−1^ (Figure 7b) display symmetric triangular profiles, showing excellent Coulombic efficiency of the NHCS-30 based symmetric supercapacitors. The specific capacitance was 158.1 F·g^−1^ at 0.5 A·g^−1^ and slightly decreased to 137.5 F·g^−1^ at 10 A·g^−1^, demonstrating the good rate performance of the symmetric supercapacitor (Figure 7c). The retention rate of the specific capacitor can still maintain 86.4% after 2000 cycles in 2 A·g^−1^, with outstanding electrochemical stability (Figure 7d). Moreover, the Nyquist plots measured from the first and the 2000th cycle (Figure 7e) also indicate that the high electrode conductivity did not change during the test. The Ragone diagram of the NHCS-30 based symmetric supercapacitors is shown in Figure 7f. It can be seen that the highest energy density of 14.1 Wh·kg^−1^ is exhibited at a power density of 200 W·kg^−1^, while the energy density still reaches 12.2 Wh·kg^−1^ at a power density of 4000 W·kg^−1^.

## 4. Conclusions

A novel self-template strategy was developed to convert resorcinol and formaldehyde resin into nitrogen-doped hollow carbon nanospheres, mainly involving the selective dissolution of acetone, an orientation arrangement of CTAB, deprotonation of AFC and followed by a carbonization process. The introduction of acetone and CTAB is beneficial in forming a multiporous structure. Meanwhile, the deprotonation of AFC is the core step to adjust the porosity and element composition. The introduction of only 30 mg AFC as ingredient significantly enhanced the SAA (1987 m^2^·g^−1^), mesopore proportion (98%), graphitization (*I*_D_/*I*_G_ = 1.02), and the contents of heteroatoms of the subsequent carbonization products. The NHCS-30 sample was utilized as a supercapacitor electrode and processed high specific capacitance of 206.5 F·g^−1^ at 0.2 A·g^−1^, with a good rate capability of 125 F·g^−1^ at 20 A·g^−1^ as well as outstanding electrochemical stability (85% retention after 5000 cycles) in a 6 M KOH electrolyte. The assembled NHCS-30 based symmetric supercapacitor performed stably, with energy density of 14.1 Wh kg^−1^ at a power density of 200 W·kg^−1^ in a 6 M KOH electrolyte, demonstrating that the self-template strategy is an effective means of developing functionality electrode materials while the as-prepared NHCS-30 was shown to be a potential candidate for energy storage devices.

## Figures and Tables

**Figure 1 materials-14-03619-f001:**
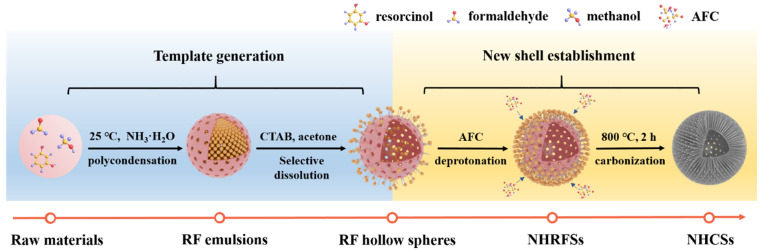
Schematic illustration of the synthetic strategy for NHCS-*m*.

**Figure 2 materials-14-03619-f002:**
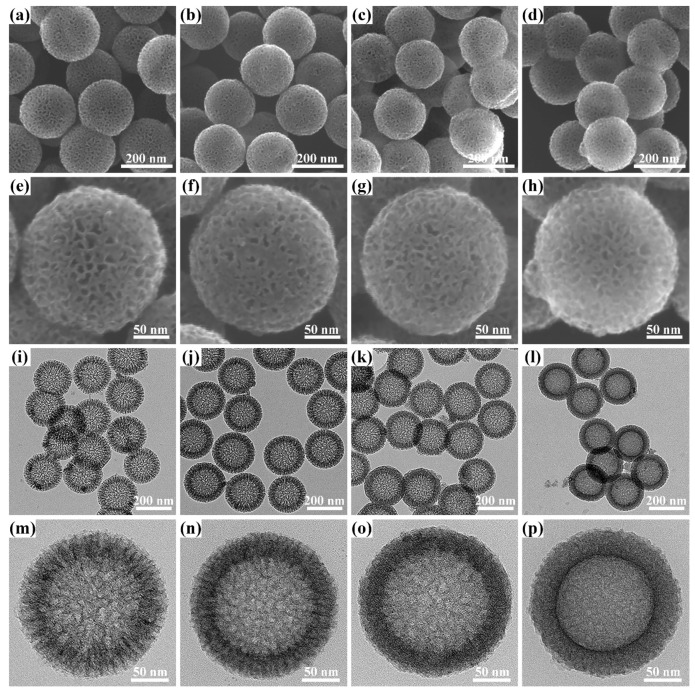
FE-SEM images of NHCS-*m* at (**a**–**d**) low-magnification and (**e**–**h**) high-magnification; TEM images of NHCS-*m* at (**i**–**l**) low-magnification and (**m**–**p**) high-magnification. (**a**,**e**,**i**,**m**) NHCS-0; (**b**,**f**,**j**,**n**) NHCS-10; (**c**,**g**,**k**,**o**) NHCS-30; and (**d**,**h**,**l**,**p**) NHCS-50.

**Figure 3 materials-14-03619-f003:**
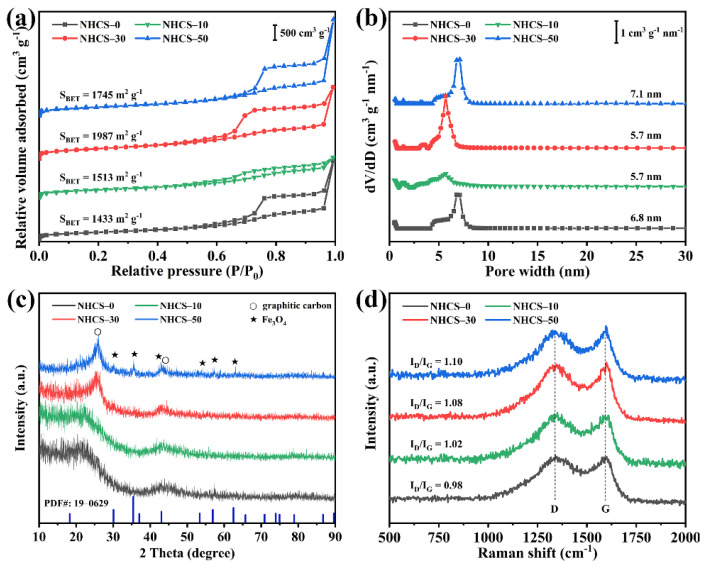
(**a**) Nitrogen adsorption-desorption isotherms and (**b**) the corresponding DFT pore size distribution curves; (**c**) XRD patterns; (**d**) Raman spectra of NHCS-*m*.

**Figure 4 materials-14-03619-f004:**
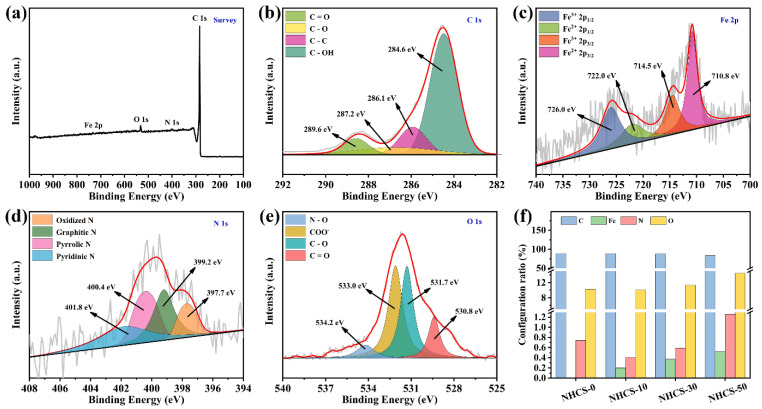
(**a**) XPS survey spectrum and (**b**–**e**) high-resolution C 1s, Fe 2p, N 1s, and O 1s regions, respectively, of NHCS- 30; (**f**) Histogram of relative contents for different elements.

**Figure 5 materials-14-03619-f005:**
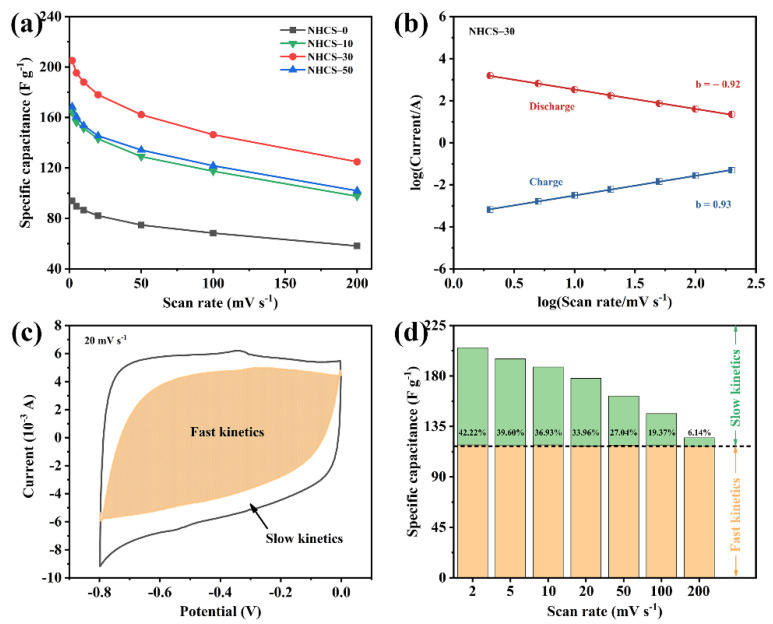
(**a**) Specific capacitance of NHCS-*m* at different scan rates; (**b**) Relationship between log(current) and log(scan rate) in the charge/discharge process for NHCS-30; (**c**) Decoupling of the capacity contributed by the fast kinetic process (shadow); (**d**) Histograms of capacitive contribution ratio for NHCS-30 at various scan rates.

**Figure 6 materials-14-03619-f006:**
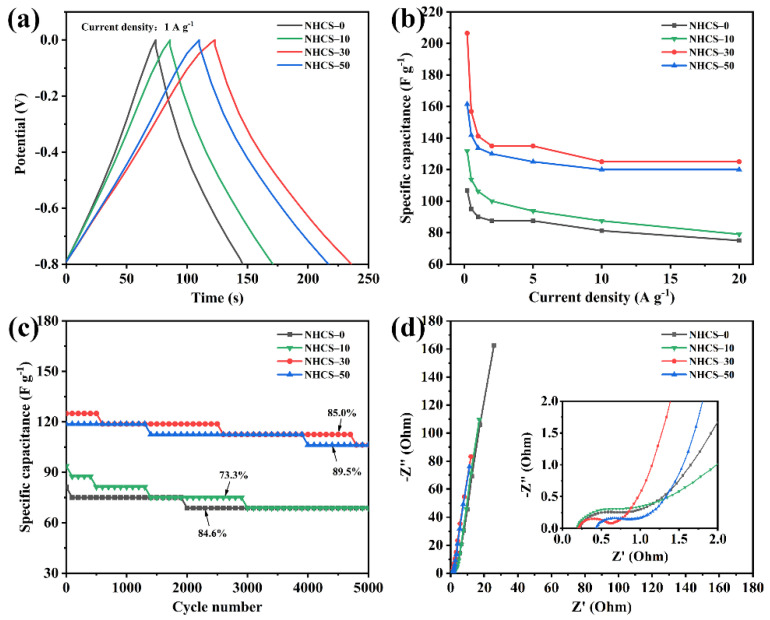
(**a**) GCD curves of the NHCS-*m* electrodes at a fixed current density 1 A·g^−1^; (**b**) Specific capacitance of NHCS-*m* calculated by different current densities; (**c**) Cycling stability of NHCS-*m* measured at 10 A·g^−1^; (**d**) Nyquist plots of NHCS-*m*.

**Figure 7 materials-14-03619-f007:**
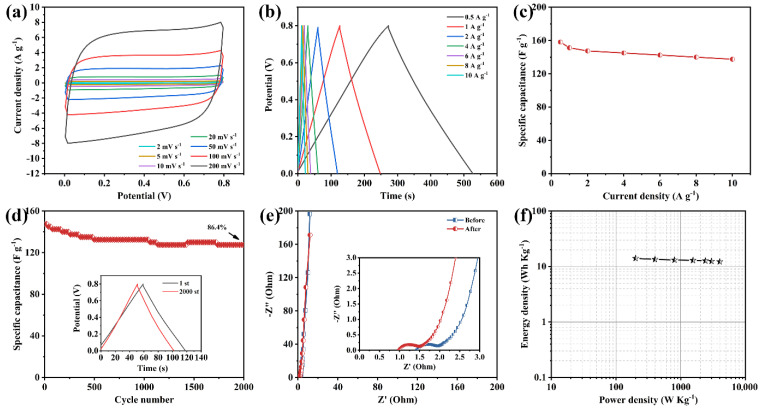
Electrochemical behaviors of the NHCS-30 based symmetric supercapacitor. (**a**) CV curves at different scan rates at voltages from 0 to 0.8 V; (**b**) GCD curves at different current densities; (**c**) Specific capacitance calculated by different current densities; (**d**) Cycling stability measured at 2 A·g^−1^; (**e**) Nyquist plots; (**f**) Energy density vs. power density.

**Table 1 materials-14-03619-t001:** Specific surface area (SSA) and pore structure parameters of NHCS-*m*.

Sample	SSA (m^2^·g^−1^)				Pore Volume (cm^3^·g^−1^)		
	*S* _BET_	*S* _mic_	*S* _mes_ ^1^	*S*_mes_/*S*_BET_ (%)	*V* _pore_	*V* _mic_	*V*_mic_/*V*_pore_ (%)
NHCS-0	1433	169.5	1263.5	88.2	5.370	0.079	1.5
NHCS-10	1513	162.0	1351.0	89.3	4.375	0.078	1.8
NHCS-30	1987	178.6	1808.4	91.0	4.056	0.081	2.0
NHCS-50	1745	176.9	1568.1	89.9	2.330	0.074	3.2

^1^ Mesoporous surface area (*S*_mes_) = *S*_BET_ − *S*_mic._

## Data Availability

The data presented in this study are available on request from the corresponding author. The data are not publicly available due to funder data retention policies.

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
