# Peer review of "Self-Template Synthesis of Nitrogen-Doped Hollow Carbon Nanospheres with Rational Mesoporosity for Efficient Supercapacitors"

_materials, 2021, doi:10.3390/ma14133619_

Round 1

Reviewer 1 Report

  1. The summarize of manuscript content:
    The paper describes a simple and effective method for producing nitrogen-doped hollow carbon nanospheres based on available compounds (resorcinol and formaldehyde). Methods for studying the obtained nanoparticles are presented and the electrochemical properties manifested by them are shown as potential energy storage devices.
  1. The strengths of the work are a clearly written introduction with a designation of potential practical applications, a description of the experimental part.
    Of the shortcomings, it is worth noting the complex language of writing the work, which greatly complicates the acquaintance with it. Also, as already noted in the review, the authors do not provide a full analysis of the data obtained in the electrochemical experiment, but only state the obtained values in fact. Finally, the criteria for the choice of starting materials are also not described (for more details, see the review).
  1. The major points for the improvement:
    a. The text of the work needs to be simplified in general, at the moment it looks like a representation of a large set of numbers and values, some of which do not need a strict presence in the manuscript (can be transferred to supplementary materials for example or so). This concerns only a part of the results and their discussion.
    b. Indicate the advantages of initial compounds used in the work for obtaining of nitrogen-doped hollow carbon nanospheres in comparison with many other known approaches (which also use simple and affordable substances)? It means that all compounds are, in fact, simple and affordable, as are the methods for obtaining nanoparticles from them, but how does the described approach stand out?
    c. Provide an objective comparative analysis of the obtained NHCSs with different magnification with those already studied, do their effectiveness differ or does it remain at the same level?

Reviewer 2 Report

The manuscript entitled “Self-template synthesis of nitrogen-doped hollow carbon nanospheres with rational mesoporosity for efficient supercapacitors” Authors present synthesis stages to obtain nitrogen-doped carbon nanospheres based on the conversion of resorcinol and formaldehyde and their characterization by SEM, sorption of nitrogen, Raman spectroscopy and XPS. Their activity in alkaline medium was characterized by galvanostatic charge/discharge measurements.

The discussions are well structured and experimental results prove conclusions of this work.

Page 7, Table 1, the number of SSA should be integer number without decimal point.

Page 10, line 296, should be with space “were calculated”.

I recommend description of the advantages of the used method in comparison with the known in literature. Also I recommend comparison of electrochemical properties with the known similar materials described in literature.

I recommend this article to be published in Metals after minor revision.

Reviewer 3 Report

The paper by Zhang, Sun and co-workers deals with the synthesis of nitrogen-doped hollow carbon nanospheres (NHCS) to be exploited as supercapacitors materials. The topic is appropriate for the journal and could be interesting for the journal readership. I therefore recommend publication after the following points will be properly addressed by authors:

- the use of ammonium citrate to prepare N-doped carbon nanomaterials has some precedents in literature that should be cited. See for example 10.1039/c5cc05259a. In particular in this latter paper the role of the ammonium citrate amount on the final material properties are discussed as well as done here by authors.

- the XPS component at 399.2 eV is attributed by authors to pyrrolic nitrogen. However, the BE seems too low for this type of N species, are they sure that it is not due to Fe-N species? See for example 10.1039/c9ee03027a

- as authors well illustrate in the introduction, other NHCS materials have been previously reported for their exploitation as supercapacitors. A comparison of the presented performance with that of previously reported related materials should be useful to better circumscribe the reported materials performance

Round 2

Reviewer 1 Report

For some reason, the authors made only minor changes to the text of the article, so a new reference was added, the parameters in the table were corrected and one inaccuracy in the text was corrected. None of the points of the previous review were taken into account. The article should be returned for revision.

Reviewer 3 Report

Even if I don't fully agree with your replies, I suggest the publication of the partially revised manuscript

Author Response

Thank you for your comments! However, in this round review, the reviewer suggested publication and did not provide the specific revision comments. Thus, I just did the English check again in this round.